# Alloying Effect of Silver on Zirconia Support Manipulated Palladium Catalyst for Methane Combustion

**Mengmeng Chu, Fan Cai, Xiaohong Cao, Li Xing, Lefu Yang \*, Xiaodong Yi and Weiping Fang**

National Engineering Laboratory for Green Chemical Productions of Alcohols-Ethers-Esters, College of Chemistry and Chemical Engineering, Xiamen University, Xiamen 361005, China; mmchu@139.com (M.C.); caif@xmu.edu.cn (F.C.); cxhongcassie@126.com (X.C.); lixing13959265991@163.com (L.X.); xdyi@xmu.edu.cn (X.Y.); wpfang@xmu.edu.cn (W.F.)

\* Correspondence: lfyang@xmu.edu.cn

**Abstract:** PdAg/ZrO$_2$ alloy catalysts calcined at different temperatures were employed to elucidate the effect of support-metal interaction (SMI) on methane combustion. Combustion activity was depressed when the sample was calcined at an elevated temperature from 500 °C to 700 °C. However, calcination at 850 °C enhanced the beneficial SMI, which facilitated a more active phase for the oxidation reaction. The high-resolution transmission electron microscopy experiments show that a special micro-domain structure at the interface is formed during the reduction pretreatment. H$_2$-TPR and O$_2$-TPD measurements illustrate that the active phase would undergo reconstruction upon redox cycles. The active phase manipulated by the support is more suitable for combustion reaction in the course of temperature altering.

**Keywords:** alloy catalyst; support−metal interaction; PdAg alloy; micro-domain; methane combustion

## 1. Introduction

Prominent environmental problems, especially the greenhouse effect, are increasingly being caused by unburned methane [1–4] and large amounts of CO$_2$ emissions [5]. Catalytic combustion is a promising technology that can convert methane into carbon dioxide at relatively low temperature and has been extensively studied in recent years. The supported Pd-based catalyst has excellent catalytic performance [6–9], but the Pd-based catalyst has many shortcomings, impoverished thermal stability [10–12], poor selectivity [13], needy resistance to sulfide poisoning [14], etc. However, alloy catalysts can solve these problems. Pd-Pt alloy catalysts have good high-temperature thermal stability [15,16] and enhance the anti-poisoning performance [17], Pd-Ni alloy catalysts have an adjustable electronic structure showing higher power density and high stability [18], and Pd-Ag nanoalloy catalysts can increase selectivity [13] and exhibit excellent activity [19,20]; thus alloy catalysts are increasingly gaining value.

A number of transition metal oxides supported on oxide from Groups II A, III B, IV B, and V B exhibit strong SMI, as reported by Tauster et al. in 1978 [21]. Murata et al. [22] illustrated that structure and particle size of the Pd nanoparticles were influenced by the SMI. Using H$_2$ and CO chemisorption on metal particles supported catalysts. O'Shea et al. [23] found that the SMI effect could be explained via electron transfer from the support to the metal (electronic factor), or by the formation of intermetallic phases after reduction at high temperatures. As a consequence, the SMI [21–25] plays a decisive role in the structure and dispersion of the active phase.

In this study, experimental evidence demonstrates a relationship between the above-mentioned phenomena through methane combustion, H$_2$ temperature programmed reduction (H$_2$-TPR), and O$_2$ temperature programmed desorption (O$_2$-TPD) measurements. For this purpose, PdAg/ZO$_2$ catalysts

are employed, and the calcining temperature is used as the main experimental variable. It is expected that the obtained results can be applied as the fundamental foundation for the catalytic methane combustion process, especially for the SMI and structure of the active phase at low temperatures.

## 2. Results and Discussion

### 2.1. Microstructure of Catalyst

2.1.1. X-ray Diffraction Analysis

Figure 1 shows the XRD patterns of Pd/Z and PdAg/Z series alloy catalysts; the crystal structure of $ZrO_2$ is in monoclinic phase (JCPDS 01–086–1451), and the sharp peaks at $2\theta$ = 24.5°, 28.2° and 31.6° can be assigned to (110), (−111) and (111) planes, respectively. We see that the dominant plane of $ZrO_2$ is the (−111) plane. After co-loading, the Pd and Ag, PdAg/Z series alloy catalysts prepared by wetness impregnation method exhibit the same crystal structure as that of $ZrO_2$. Importantly, no diffraction peaks ascribed to any Pd and Ag species can be observed, indicating that the particle size of Pd or Ag is quite small. The reasons may be: (I) the loading amount metal is relatively low; (II) the SMI can promote the dispersion of PdAg on $ZrO_2$ surface and restrain aggregation.

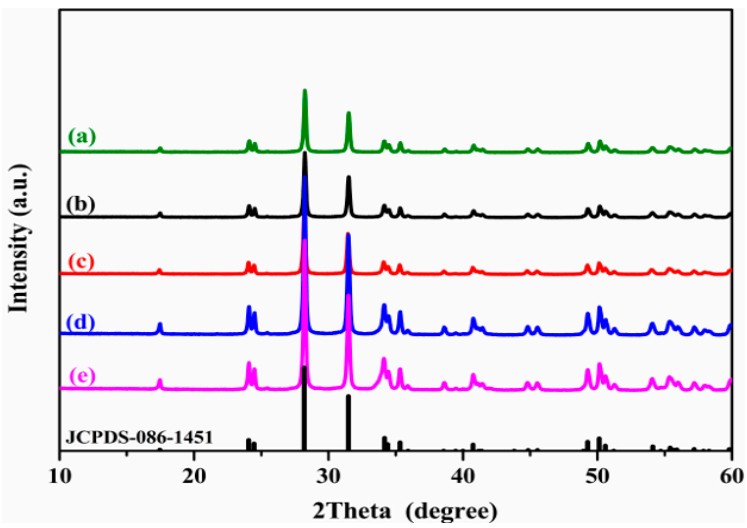

**Figure 1.** X-ray diffraction patterns of the support and catalysts: (**a**) m-$ZrO_2$; (**b**) 1% Pd/Z; (**c**) 1% PdAg/Z; (**d**) 2% PdAg/Z; (**e**) 5% PdAg/Z.

2.1.2. High-Resolution Transmission Electron Microscopy Analysis

Figure 2a,b show the HRTEM images of PdAg/Z-500, PdAg/Z-850 catalysts. It can be found that although the PdAg alloy particles become bigger after calcination at 850 °C, PdAg can still maintain dispersity, which is consistent with the fact that the PdAg alloy could enhance the thermal stability of Pd/Z catalysts [20]. Figure 2c,d illustrate the HRTEM images of PdAg/Z-500 that was re-oxidized under air and re-reduced under $H_2$ at 500 °C. Clearly, the PdAg alloy particles are almost flat on the $ZrO_2$ surface after re-oxidation. This phenomenon illustrates that the degree of SMI was strengthened after re-oxidation. Further, the SMI effect inhibits the growth of the metal particles at the interface between PdAg and the $ZrO_2$ surface, thus forming a special micro-domain structure at the interface during the reduction process. By calculating the lattice fringes, we find that the SMI could manipulate the structure of the active phase. The fringe of the alloy particle d = 0.2296 nm is Pd (111) plane (PDF # 87–0637, d = 0.2301 nm), for the fringe statistics with d = 0.2700 nm at the micro-domain structure is t-PdO (002) plane (PDF # 43–1024, d = 0.2668 nm). Additionally, the d = 0.3200 nm is ascribed to the m-$ZrO_2$ (−111) plane (PDF # 86–1451, d = 0.3164 nm). Therefore, it can be concluded that the special micro-domain structure at the interface was a specific manifestation of SMI.

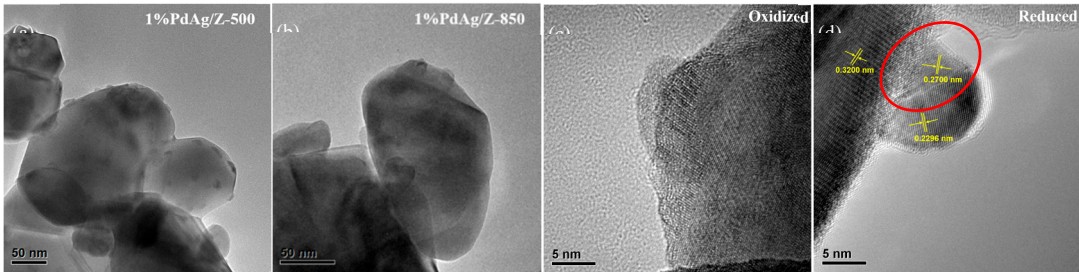

**Figure 2.** HRTEM images of supported alloy catalysts: (**a**) PdAg/Z-500; (**b**) PdAg/Z-850; (**c**) PdAg/Z-500 were re-oxidized under air at 500 °C; (**d**) $H_2$ re-reduced at 500 °C.

Huang et al. [26] found that the ability of transforming Pd to PdO can show excellent performance for methane combustion. Besides, due to the good interaction between Pd and support, the mobility of active oxygen from support to Pd can readily achieve the goal of facilitating the stability of PdO species. In addition, Nilssona et al. [27] used XAFS characterization and found that the oxidation of Pd on the surface and bulk have a different impact on methane combustion. In the beginning of the reaction period, the surface of the catalyst is oxidized to form PdO. Following the reaction, the Pd oxidation state continues to increase, which corresponds to bulk oxidation of Pd. This indicates that there is an interface between the surface PdO and bulk PdO. From Figure 2c,d, after re-oxidation, the catalyst is pure PdO. Due to the SMI, the interface is still PdO after re-reduction. Jiang et al. [28] reported that the bimetallic Pt–Pd/$ZrO_2$ catalysts can maintain smaller nanoalloy particle size and narrow size distribution for alloy catalysts. As for PdAg/Z catalysts, due to the SMI, the metallic Pd and oxide PdO are reconstructed during the pretreatment, which makes the catalyst surface more enriched with PdO.

*2.2. Effect of Calcination Temperature*

2.2.1. Catalytic Activity for Methane Oxidation

To investigate the SMI effect on the catalytic performance, the PdAg/Z alloy catalysts prepared by different calcining temperature were evaluated by methane combustion. As shown in Figure 3, PdAg/Z-500 ignites the oxidation reaction at 331 °C ($T_{10}$). After the ignition, the methane conversion increases rapidly with reaction temperature, and the burn-off is obtained at 443 °C ($T_{90}$). As expected, the calcination at higher temperature results in an obvious deactivation, which could be related to the sintering of active phase. The sample PdAg/Z-600 cannot ignite the oxidation until about 140 °C higher than PdAg/Z-500; even the burn-off still remains unachieved at 600 °C. However, further calcination at 700 °C brings about the recovery of catalytic activity and the light-off temperature over PdAg/Z-500 drops down to 423 °C. This implies that the construction towards a beneficial active phase occurs, much of it driven by the moderate SMI from the thermal treatment. The positive effect of SMI is proven to be enhanced after the supported alloy catalyst is calcined at 850 °C. PdAg/Z-850 is able to ignite methane oxidation at 370 °C.

Since the sintering at high temperature is usually irreversible, the recovery of catalytic activity cannot be attributed to the dispersion of active metal component. Because the methane oxidation promoted by noble metal catalysts typically has structural sensitivity, it will probably take advantage of active phase reassembly driven by SMI on the support template. This assumption has also been verified by an aging test over PdAg/Z-850, in which the active phase is optimized under reaction atmosphere. As shown in Figure 4, the catalyst was continuously tested at 350 °C for 50 h. By holding the temperature at 350 °C, the methane conversion gradually rose from 10 to 77% in the initial 6 h until a stable conversion of 80% was attained. In the whole period of 50 h aging test, no temporary or inclining decrease was observed. As the catalytic methane oxidation is composed by cyclic reduction and oxidation of the active phase, it offers the active phase an opportunity to reconstruct on the support template. The form of support template and interaction between the active phase and support would

inevitably play a part in the reconstruction. The above experiments demonstrate that the zirconia support with monoclinic crystal phase and the epitaxy SMI are suitable for the improvement of active phase for catalytic methane combustion.

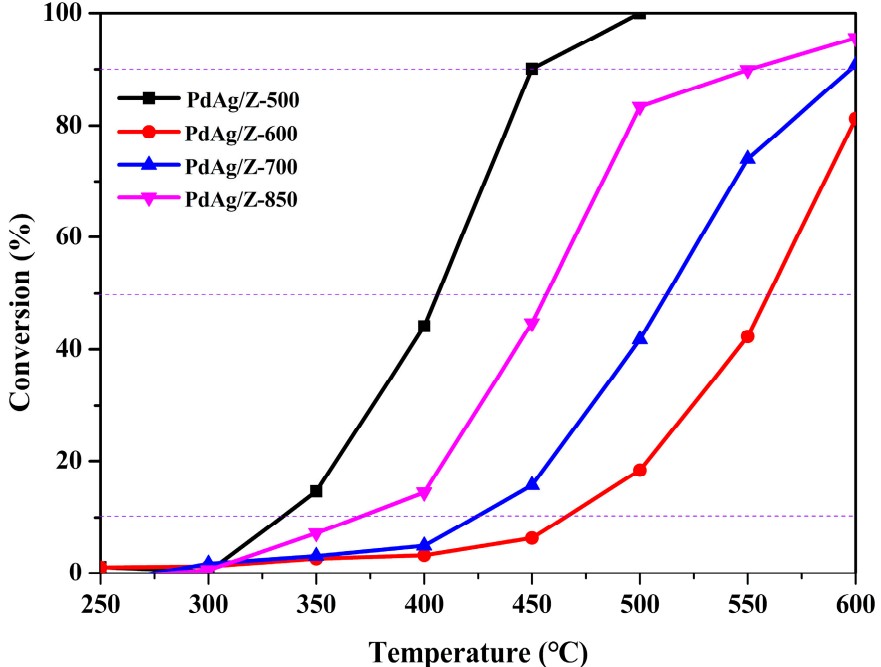

**Figure 3.** Methane conversion curve by PdAg/Z alloy catalysts calcined at different temperatures. The reaction conditions were: heating rate 10 °C min$^{-1}$ and flow rate of the reacting feedstock 40 mL min$^{-1}$ (2 vol % CH$_4$/Ar and 4 vol % O$_2$/Ar, space velocity 72,000 h$^{-1}$).

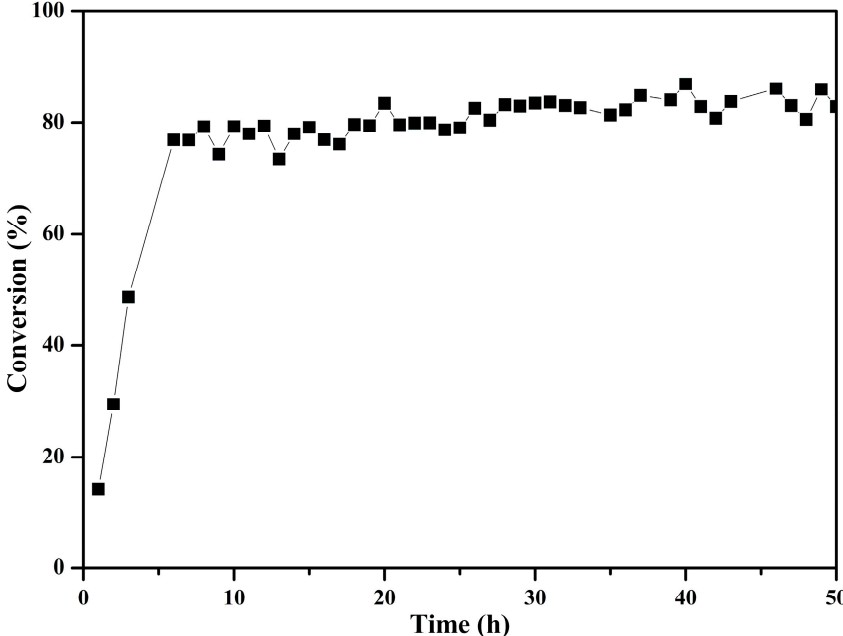

**Figure 4.** Methane conversion of the aging test for the PdAg/Z-500 catalyst at 350 °C for 50 h. The feed gas consisting of CH$_4$:O$_2$ = 1:2, GHSV = 72,000 h$^{-1}$.

### 2.2.2. $O_2$ Temperature Programmed Desorption Analysis

The catalytic activity is highly related to the active oxygen species; thus, the $O_2$-TPD experiment was performed on the catalysts (Figure 5). The figure shows that $ZrO_2$ does not have any $O_2$ desorption signal, which means that all $O_2$ desorption is caused by the decomposition of Pd and Ag oxides and the migration of $O_2$ between the PdAg and $ZrO_2$ interface [29]. When the calcination temperature is lower than 700 °C, the desorption temperature of $O_2$ gradually increases from 661.8 °C to 698.2 °C as the calcination temperature rises. However, when the calcination temperature is 850 °C, the desorption temperature of $O_2$ drops to 687.8 °C. This is because when the calcination temperature is lower than 700 °C, the active phase sinters as the calcination temperature increases, strengthening the Pd-O-Zr structure and making the surface oxygen species more stable. However, when the temperature is up to 850 °C, it enhances the moderate SMI. It facilitates the active phase, which promotes active oxygen migration and reduces the desorption temperature of $O_2$. Combining the catalytic activity and $O_2$-TPD characterization, the catalytic activity is consistent with the change trend of $O_2$ desorption temperature.

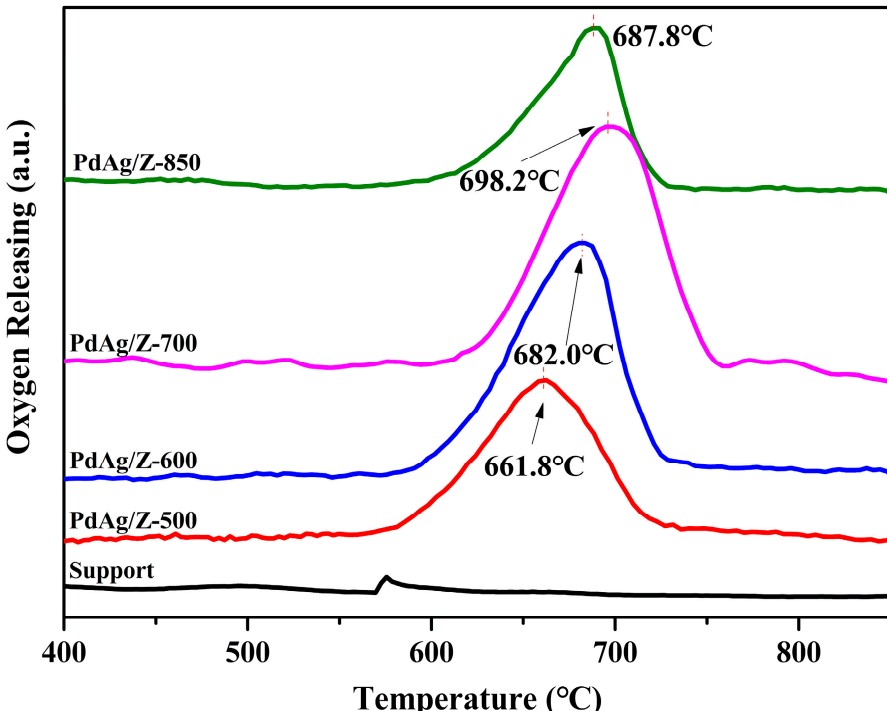

**Figure 5.** $O_2$ temperature programmed desorption ($O_2$-TPD) profiles of PdAg/Z catalysts calcined at different temperatures.

To explore the reconstruction behavior of active oxygen during the reaction, the $O_2$-TPD cycle desorption experiment was designed; the results are shown in Figure 6. After the oxygen desorption temperature was elevated to 700 °C, it was immediately dropped to 200 °C and then increased again. When the temperature rises to 574 °C again, $O_2$ desorption occurs as before. This implies that by the cycling of temperature, $O_2$ with desorption temperature higher than 700 °C will be reconstructed to low temperature for desorption. It is suggested that the manipulation of the PdO active phase by m-$ZrO_2$ template is more advantageous to the exchange of oxygen.

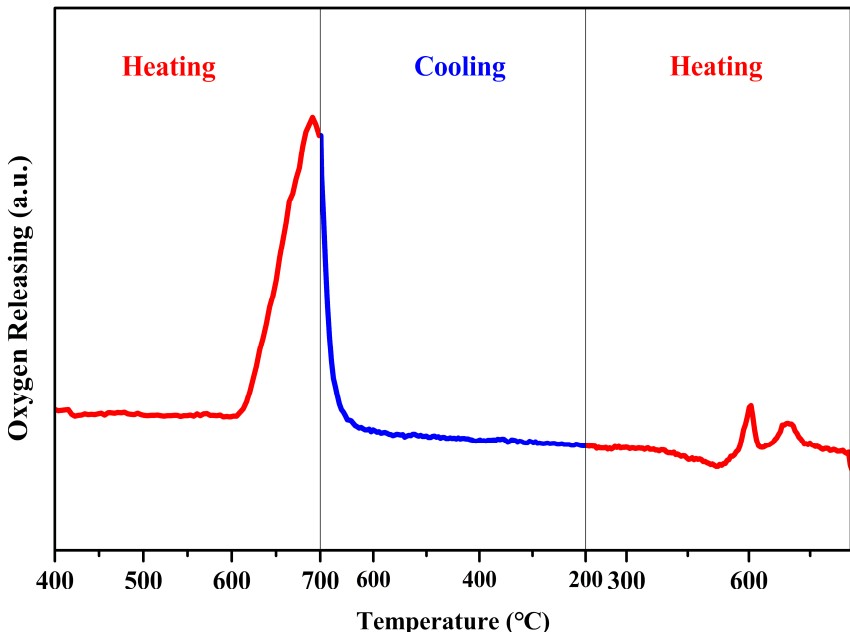

**Figure 6.** $O_2$ temperature programmed desorption ($O_2$-TPD) cyclic desorption profiles of the PdAg/Z-500 catalyst.

### 2.2.3. $H_2$ Temperature Programmed Reduction Analysis

Figure 7 shows the $H_2$-TPR patterns of PdAg/Z catalysts. In order to better observe the hydrogen consumption signal, the polarity of the detector was reversed during the $H_2$-TPR experiment, such that the $H_2$ consumption signal was converted into $H_2$ desorption signal. The inner illustration is the detection limit of the detector. The $H_2$-TPR profiles consist of three reduced peak, $\alpha$, $\beta$, and $\gamma$. $\alpha$ (about 50 °C) is attributed to the reduction of surface PdO species and absorbed oxygen on the surface of catalyst [30]. $\beta$ (80–120 °C) is pertinent to the decomposition of $PdH_X$. $\gamma$ (T max 250–350 °C) is assigned to the reduction of subsurface PdO on the catalyst surface [31]. The decomposition temperature of the peak reflects the SMI. The PdAg/Z alloy catalyst calcined at 500 °C had the highest decomposition temperature of $PdH_X$ at 81.7 °C, but with the increase of calcination temperature, $PdH_X$ decomposition temperature is gradually reduced, that is because with the increase of calcination temperature, the degree of SMI is strengthened and free Pd on the catalyst surface is reduced, making $PdH_X$ to be too stable to decompose at lower temperatures.

### 2.2.4. CO Temperature Programmed Reduction Analysis

Compared with $H_2$-TPR, CO-TPR can avoid the split of $H_2$. Besides, the reduction of CO is weaker and the reduction slower than $H_2$, so it can well express the relationship between the various oxide species. Figure 8 is the CO-TPR curve of a series of PdAg/Z catalysts. As shown in Figure 8, the reduction of PdAg/Z bimetal has mainly three peaks. For the $ZrO_2$ calcined at 500 °C, the $CO_2$ peak at 425 °C is attributed to the reduction of the lattice oxygen and active oxygen on the support surface. Compared to the PdAg/Z catalysts, the reduction temperature decreased because the SMI was able to promote the migration of lattice oxygen and facilitate the oxidation of CO [32]. Changing of the calcination temperature enabled the alteration of the structure of Pd-O-Zr, and then changed the degree of manipulation of PdAg by $ZrO_2$. From Figure 8, it is found that the $\alpha$ peaks (207–232 °C) become more and more sharp as the calcination temperature increased, proving that as the calcination temperature increased, the degree of SMI gradually strengthened, enabling the active oxygen species to easily migrate on the catalyst surface. However, as the calcination temperature (<850 °C) cannot produce moderate SMI, the PdAg/Z-850 would have better catalytic performance than PdAg/Z-600 and PdAg/Z-700. The $\beta$ peaks at 390–430 °C are the reduction of lattice oxygen and active oxygen

at the interface between Pd and $ZrO_2$. This is the main active component of the PdAg/Z catalyst. From the CO-TPR curves, we find that the reduction temperature of the lattice oxygen and the oxygen species at the support-metal interface is consistent with the change trend of the catalyst performance, which indicates that the reducibility of active oxygen at the support-metal interface and the mobility of lattice oxygen would directly affect the catalytic performance. This is consistent with the $H_2$-TPR results.

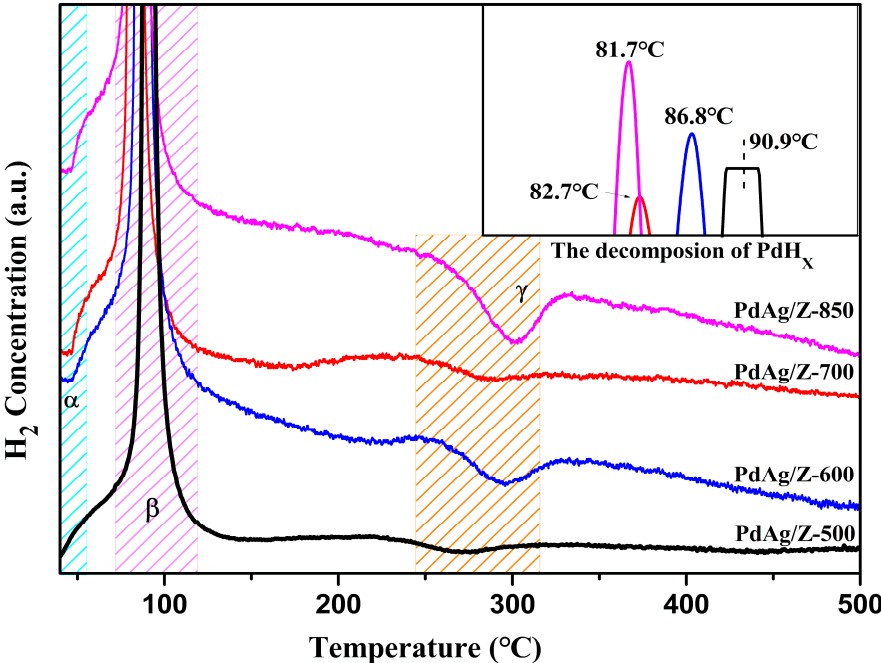

**Figure 7.** $H_2$ temperature programmed reduction ($H_2$-TPR) profiles of PdAg/Z catalysts calcined at different temperatures.

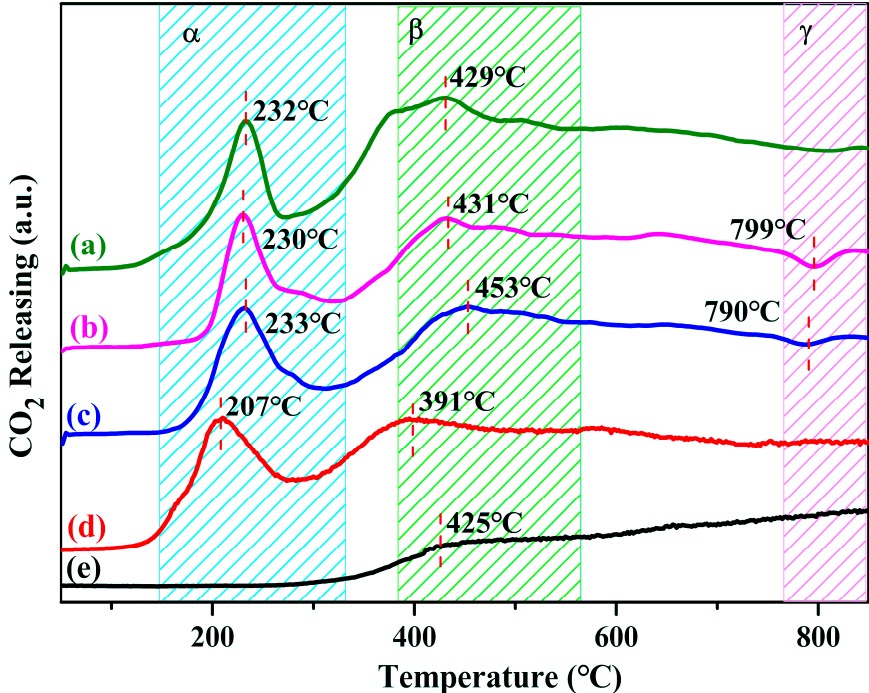

**Figure 8.** CO temperature programmed reduction (CO-TPR) profiles of PdAg/Z catalysts calcined at different temperatures. (**a**) PdAg/Z-850; (**b**) PdAg/Z-700; (**c**) PdAg/Z-600; (**d**) PdAg/Z-500, and (**e**) support.

### 2.2.5. X-ray Photoelectron Spectroscopy Analysis

Figure 9 shows the XPS curves of PdAg/Z alloy catalysts calcined at different temperatures. The figure displays that the binding energy of $Pd^{2+}$ remains basically unchanged at 336.5 eV with the increase of calcining temperature [33,34], but the binding energy of metallic Pd shifts from 335.0 to 335.3 eV [35]. It means that the content of metallic Pd decreased. However, combined with Table 1, it can be found that the binding energy of Ag reduced 0.2 eV when the calcination temperature was at 850 °C, which shows that the content of Ag increased. Therefore, the addition of Ag could maintain the PdO-Pd mixture active phase containing much more $Pd^{2+}$ at a high temperature and thus the catalytic activity is improved. The statistical catalyst surface atomic ratio of each element is listed in Table 1. With the increase of calcination temperature, the Pd/Zr ratio remains the same. This shows that the PdAg/Z-850 catalysts could still maintain good dispersion; thus, Ag can improve catalyst high temperature sintering resistance. It can be seen from the ratio of $Pd^{2+}/Pd^{0}$ that as the calcination temperature increased, the change trend of the ratio of $Pd^{2+}/Pd^{0}$ was consistent with that of the catalytic activity. It is indicated that the active phase with a higher $Pd^{2+}/Pd^{0}$ ratio will have a better catalytic activity for methane combustion.

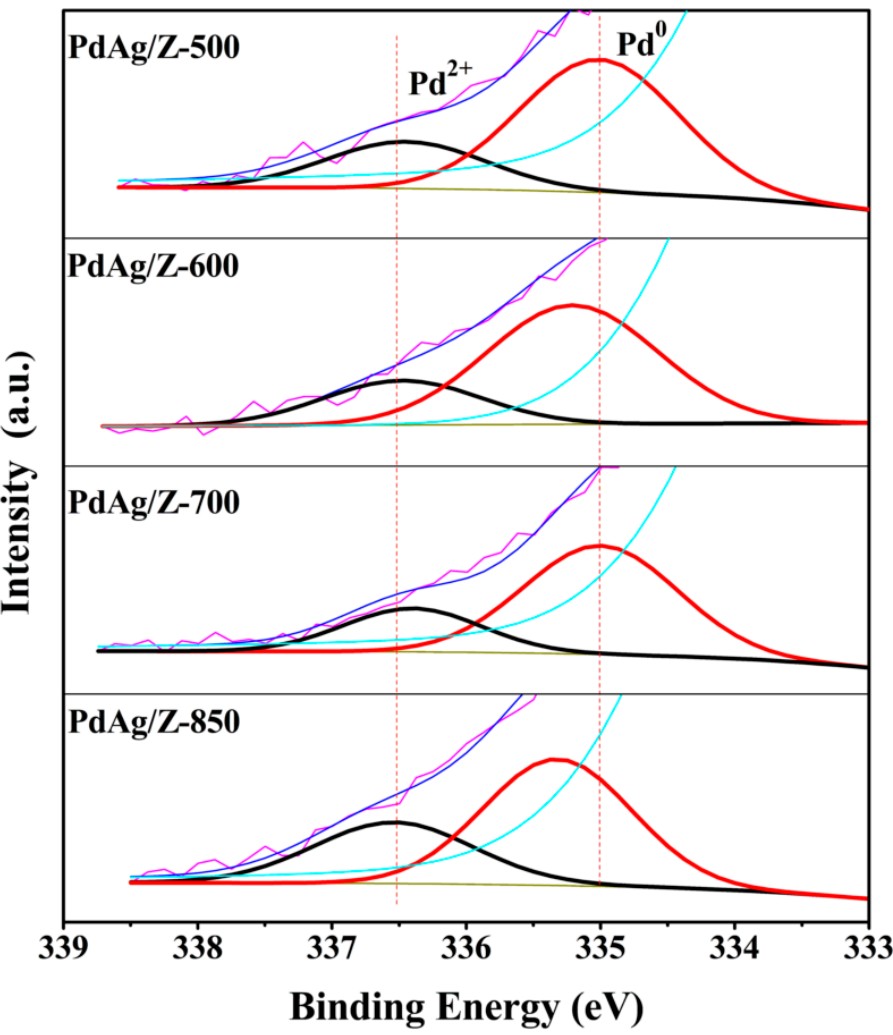

**Figure 9.** XPS curves of PdAg/Z alloy catalysts calcined at different temperatures.

**Table 1.** Chemical state analysis from XPS characterization.

| Calcined Temperature | Peaks of Binding Energy / eV | | | | | Atomic Ratio | | |
| | Pd $3d_{5/2}$ | | Ag $3d_{5/2}$ | | | | | |
| /°C | Pd | $Pd^{2+}$ | Ag | $Ag^+$ | O 1s | Pd/Zr | $Pd^{2+}$/Pd | $Ag/Ag^+$ |
|---|---|---|---|---|---|---|---|---|
| 500 | 335.0 | 336.5 | 367.4 | 368.2 | 529.8 | 0.08 | 0.35 | 0.53 |
| 600 | 335.2 | 336.5 | 367.4 | 368.0 | 529.8 | 0.12 | 0.34 | 0.71 |
| 700 | 335.1 | 336.4 | 367.4 | 368.2 | 529.9 | 0.09 | 0.35 | 2.67 |
| 850 | 335.3 | 336.5 | 367.2 | 368.4 | 529.9 | 0.08 | 0.53 | 5.10 |

## 3. Materials and Methods

### 3.1. Catalysts Preparation

A reference catalyst of 1 wt% PdAg/ $ZrO_2$ was prepared by the incipient wetness impregnation method. The Pd:Ag molar ratios of the finished alloy catalyst was 10:1. $Pd(NH_3)_4(NO_3)_2$ (Strem Chemicals, Newburyport, MA, USA, 5.0 wt.% Pd) and $AgNO_3$ (XiLong Chemicals, Shantou, Guangdong, China, 99.8 %) were dissolved in the deionized water, then $ZrO_2$ (J&K Chemicals, Beijing, China, 99.5 %) powder was added into the mixture solution under slow stirring. After impregnation, the sample was dried at 120 °C overnight. In order to explore the effect of calcination temperature on SMI, the catalysts were calcined at different temperature after 7 days' ripening. A series of PdAg/Z catalysts calcined at 500, 600, 700, and 850 °C were named PdAg/Z-500, PdAg/Z-600, PdAg/Z-700, and PdAg/Z-850 respectively.

The procedure of seven-day ripening is introduced to settle the unsteady mixed metal components, which were freshly deposited on the zirconia support. As shown in the supporting information, the sample without ripening usually presents a more oxidative state after calcination. The process of ripening can limit the SMI to the interface between the metal and the support, on which further interaction is suppressed by particle growing. Given that the calcination intervals usually last for a few days, the conformance of samples would be guaranteed by adequate ripening.

### 3.2. Catalytic Activity for Methane Oxidation

Catalytic methane combustion was performed in a quartz fix-bed flow reactor at atmospheric pressure. The gas flow was controlled by a mass flow controller (Sevenstar D07-19B MFC, Beijing, China,) to regulate the composition of gas mixture. The reaction temperature was measured by a K-type thermocouple placed at the bottom of the catalyst bed in order to measure the exit temperature close to the reaction zone. Using 50 mg, 40–60 mesh (0.25–0.40 mm in diameter) fresh sample was loaded in the fix-bed quartz reactor ($\phi_{out}$ = 6 mm, $\phi_{in}$ = 4 mm) with heating rate of 10 °C $min^{-1}$ under a flow rate of 40 mL $min^{-1}$ of reacting feedstock (2 vol % $CH_4$/Ar and 4 vol % $O_2$/Ar, space velocity 72,000 $h^{-1}$) under stoichiometric conditions ($O_2/CH_4$ = 2) of total oxidation. The activity measurement started from 250 °C and was performed at every 50 °C of temperature elevation. When the temperature reached the programmed condition, it was maintained for 20 min under the reaction gas atmosphere. The effluents from the reactor were analyzed by on-line gas chromatograph (Haixin GC 950, Shanghai, China,) equipped with a TCD detector. Prior to TCD detection, the effluent mixture was separated by a packed column of 5A molecular sieve.

### 3.3. Catalyst Characterization

X-ray diffraction analysis of the catalysts were carried out using a Rigaku Ultima IV X-ray powder diffraction spectrometer (Japan), operated at 35 kV, and 15 mA employing Cu Ka radiation. Specifically, step width was 0.02° and scan range was measured from 20 to 80° with scanning speed of 10° per min.

High-resolution transmission electron microscopy studies were performed on Tecnai 30 (Holland). Samples were prepared by dry dispersing the catalyst powder onto a holey carbon film supported by a 300 mesh copper grid. The appearance and the chemical dispersion of Pd, Ag, and $ZrO_2$ were

recorded on transmission electron microscopy images. The fringe statistics of alloy particles and the support were calculated by Digital Micrograph software (USA).

$H_2$ temperature programmed reduction ($H_2$-TPR) experiments were carried out at Shimadzu GC 8A chromatography and the $H_2$ signal was monitored by TCD. About 50 mg catalyst particles (dp 0.250~0.425 mm) were placed in a quartz micro-reactor. Prior to the experiment, the sample was pretreated in the argon (99.99%) flow (30 mL min$^{-1}$) at 300 °C for 1 h, followed by a decrease to 40 °C. When the temperature cooled to 40 °C, the flow of 5% $H_2$/Ar was switched into the reactor. Finally, the temperature increased from 40 °C to 800 °C at a heating rate of 10 °C min$^{-1}$. During the period of temperature-rise, the concentration of $H_2$ in the effluent gas was continuously recorded by TCD.

CO temperature programmed reduction (CO-TPR) characterization was monitored by Hiden QIC-20 Mass Spectrometer (UK). A continuous flow of air passed over the 50 mg fresh sample. The temperature increased from room temperature to 300 °C at a rate of 10 °C min$^{-1}$ and was maintained for 60 min, followed by a decrease to 40 °C. When the temperature cooled to 40 °C, the flow of 2% CO/He was switched into the system. Finally, the temperature increased from 40 to 900 °C with the heating ramp 10 °C min$^{-1}$. The signal of $m/z$ = 2, 16, 18, 28, and 44 were recorded simultaneously.

$O_2$ temperature programmed desorption ($O_2$-TPD) experiments were performed using a 50 mg sample placed in a quartz reactor. The outlet of the reactor was connected with a Hiden QIC-20 quadrupole mass spectrometer. Before the $O_2$-TPD testing, the sample was oxidized in 1% $O_2$/He flow (30 mL min$^{-1}$). The sample bed was heated at a rate of 10 °C min$^{-1}$ from room temperature up to 500 °C and hold at this temperature for 30 min, then cooled to 30 °C at a rate of 5 °C min$^{-1}$. When the temperature was cooled to 30 °C, He (99.99%) was switched into the reactor. Finally, the temperature increased from 30 to 900 °C with the heating ramp 10 °C min$^{-1}$ under He (99.99%) atmosphere and signals of $m/z$ = 16, 32, 18, 28, and 44 in outlet gas were recorded.

X-ray photoelectron spectroscopy (XPS) analysis was performed on a PHI 5000C ESCA System instrument (Japan). Spectra were recorded by using monochromatized Al K$\alpha$ radiation (1486.6 V), with an X-ray power of 250 W. The spectrometer was operated in the analysis chamber: vacuum was $1.0 \times 10^8$ Torr, working voltage was 14.0 kV, and signal accumulation of 100 cycles were performed. The C 1s peak (284.6 eV) was used for the calibration of binding energy values. The $Pd^{2+}/Pd^0$ was calculated according to D = A($Pd^{2+}$)/A($Pd^0$), where A($Pd^{2+}$) is the peak area of $Pd^{2+}$ and A($Pd^0$) is that of $Pd^0$. XPS data analysis was performed with the XPSPEAK41 software (USA).

## 4. Conclusions

In this paper, a series of PdAg/Z catalysts were investigated. We find that the presence of Ag inhibits the sintering of Pd during calcination at high temperature. The HRTEM observations also reveal a specific microdomain structure between the metal and the support. This may be the evidence of SMI. In addition, it is found that the degree of SMI can be adjusted by changing the calcination temperature. The analysis of catalytic activity for methane oxidation illustrates that PdAg/Z-500 shows the highest methane conversion. As the temperature increases, the sintering of active phase occurs, which results in lower catalytic activity. However, further increase of the calcination temperature enhances the beneficial SMI. It facilitates the active phase that meets the demand of structural sensitivity for methane combustion. The PdAg/Z-850 catalyst exhibited higher activity ($T_{90}$ at 550 °C) in methane catalytic combustion. On combining the characterization results of $O_2$-TPD and TPR, the conclusion can be drawn that the degree of SMI differs with changing calcination temperature, which leads to the formation of various structures of the PdO-Pd active phase. Through XPS analysis, Ag can maintain proper $Pd^{2+}/Pd^0$; thus, the catalysts can have suitable active phase composition. Therefore, controllability of the catalytic performance could be brought about by adjusting calcining temperature.

**Supplementary Materials:** The following are available online at http://www.mdpi.com/2073-4344/10/8/863/s1, Figure S1: XPS curve of PdAg/Z alloy catalyst without ripening treatment.

**Author Contributions:** Conceptualization, M.C. and L.Y.; Data curation, M.C., F.C. and L.X.; Funding acquisition, X.Y. and W.F.; Investigation, M.C., F.C., X.C. and L.X.; Methodology, M.C. and X.C.; Project administration,

X.Y.; Software, F.C.; Supervision, L.Y. and W.F.; Writing—original draft, M.C.; Writing—review & editing, L.Y. All authors have read and agreed to the published version of the manuscript.

**Funding:** The research was funded by National Natural Science Foundation of China (Funding Number: 21373171).

**Conflicts of Interest:** The authors declare no conflict of interest.

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
