# Peer review of "Alloying Effect of Silver on Zirconia Support Manipulated Palladium Catalyst for Methane Combustion"

_catalysts, doi:10.3390/catal10080863_

Round 1
Reviewer 1 Report
The authors investigated bimetallic Pd-Ag catalysts supported on ZrO2 with regard to alloying effects in the context of methane total oxidation. By combining several profound ex situ approaches like HRTEM and XPS with temperature programmed experiments and catalytic activity tests, the authors found that the calcination temperature strongly influences the interaction between the support material and the noble metal, which was found to govern the catalyst’s activity. The study is particularly interesting from a fundamental point of view and the applied methods are suitable for comprehensively investigating the complex topics of “alloying” and “support-metal interactions”. However, the authors need to address a few issues prior to the publication.
- Could the authors please comment on how the ripening was performed (conditions) and how the process of ripening limits the SMI to the interface between the metal and the support (section 2.1)?
- In section 3.1.2 the authors discuss the particle size and claim that the SMI inhibit the growth of the metal particles. It may be suitable to give the mean particle size for the bimetallic particles of the catalyst calcined at low and high temperature, which will allow a better comparability of the catalyst samples calcined at different temperatures.
- In section 3.1.2 the authors claim that “PdAg can still maintain dispersity, which indicates that PdAg alloy could significantly enhance the thermal stability of Pd/Z catalysts”. Although this hypothesis may be right, the data presented in the manuscript do not allow drawing this conclusion. To support their assumption, the authors would need to show HRTEM images of a monometallic Pd/Z catalyst for comparison with the bimetallic PdAg/Z catalysts. I understand if HRTEM cannot be re-done for monometallic catalysts, but in this context the authors should at least cite and discuss relevant literature (several relevant studies have already been cited in the introduction) on sintering of monometallic Pd catalysts, particularly supported on ZrO2.
- In section 3.2.1 the authors claim that the stacking PdO active phase is formed along the m-ZrO2 during the reaction process. The catalytic data alone are not sufficient to conclude on such an in-depth understanding on an atomic level, but I assume ex situ HRTEM analysis after the long-term test could clearly prove this hypothesis. Has this been done? It should be explained how the authors drew their conclusion.
- The authors discuss the active species (PdO vs. Pd2+) in their PdAg catalyst, the role of alloying and the impact of the support several times. The references already cited in the introduction should be exploited also in the results and discussion section, e.g. [refs. 6, 7, 23, 33], while the inclusion of more relevant recent literature on the complex Pd-PdO system – a couple of X-ray absorption spectroscopy studies on the Pd-PdO oxidation state in the context of (pre-)reduction and (re-)oxidation have been published in 2019 and 2020 – may further increase the soundness of the discussion. In this context, also phenomena like surface roughening, particle morphology and restructuring should be discussed, since the HRTEM images presented in the paper allow valuable insight on the atomic level.
- Since methane oxidation is an exothermic reaction, the authors should specify the position of the thermocouple during catalyst testing: Was the thermocouple placed in front, after or within the catalyst bed?
- The authors highlight the benefits of alloyed Pd-based catalysts in their introduction [refs. 13-21] and I agree that all cited references are of relevance. However, I recommend differentiating a bit more, since [refs. 15-17] report on Pd-Pt catalysts and [ref. 21] reports on Pd-Ni catalysts instead of Pd-Ag alloyed catalysts.
- Abbreviations should be defined at first appearance and then should be consequently used throughout the whole paper in a uniform way.
- For the reader’s convenience, reaction conditions could be given directly in the figure caption (e.g. Fig. 3, 4).
- Since, in contrast to spectra, X-ray diffraction patterns are not strictly additive, the authors should avoid using the term “XRD spectra” (section 3.1.1) and instead consequently use the term “XRD patterns” as they do in Fig. 1.
- I recommend editing the manuscript after revision with regard to English language and style, particularly the introduction and section 3.2.2 should be improved.
- It may be a problem at my end or from the manuscript service, but the x-axis label is not properly displayed for Figs. 1, 5, 6, 7, 8, 9.
Author Response
The reply has listed in the attached file.

Reviewer 2 Report
The topic of the research is interesting and the authors have carried out an adequate amount of analysis. However I would recommend the text is edited to improve the English and experimental methods. There are a number of statements in the discussions which appear to be missing references.
I also feel it would improve the paper to broaden the HRTEM analysis to all the full range of catalysts calcined at various temperatures and the trend discussed alongside the TPR/TPD data.
Author Response
The reply is enclosed in the attached file

Round 2
Reviewer 1 Report
The authors took the feedback into account and clarified several issues. Therefore, I recommend to accept the manuscript for publication. One last minor recommendation: The authors gave a very concise answer in the author response with regard to ripening. It may be convenient for the reader to extend the corresponding paragraph in the manuscript with the information given in the answer.
Author Response
Thanks a lot for the great patience and the precise comments from reviewer1. Inspired by these suggestions, we have acquired much improvement on this manuscript.
Following the last recommendation from Mr. Reviewer1, we have added an extra paragraph in the experimental section to explain the intention to introduce the ripening step. As a comparative evidence, an XPS result of the calcined catalyst without ripening is attached as supplementary information.
Reviewer 2 Report
Many thanks for the updated manuscript. The improvements to the discussions have made the paper easier to follow. The English and detail in the method section still requires improvement. There are still minor errors in the English and some details missing in the methods. Such as temperature ramp rates and gas flow rates which would influence results if attempting to repeat the experiments.
Author Response
Thanks a lot for the great patience and the helpful advice from reviewer2.
As suggested by the advice, we have corrected several misleading expression,and all the corrections are highlighted in the manuscript.
Additionally, the details of the experiment methods are also complemented, especially in the description of H2-TPR test, the temperature ramp rate and purging condition are included.